# Prebiotics: Mechanisms and Preventive Effects in Allergy

**DOI:** 10.3390/nu11081841

**Published:** 2019-08-08

**Authors:** Carole Brosseau, Amandine Selle, Debra J. Palmer, Susan L. Prescott, Sébastien Barbarot, Marie Bodinier

**Affiliations:** 1INRA Pays de la Loire, UR1268 Biopolymers Interactions Assemblies (BIA), rue de la géraudière, BP 71627, 44316 Nantes Cedex 01, France; 2Telethon Kids Institute, University of Western Australia, 15 Hospital Ave, Nedlands, Western Australia 6009, Australia; 3School of Medicine, University of Western Australia, 35 Stirling Highway, Crawley, Western Australia 6009, Australia; 4Department of Dermatology, Nantes University Hospital, 44035 Nantes, France; 5UMR1280 Physiopathology of Nutritional Adaptations (PhAN), INRA, 44035 Nantes, France

**Keywords:** prebiotics, mechanisms, allergy, microbiota, immune system, epithelial barrier, clinical studies, preclinical studies

## Abstract

Allergic diseases now affect over 30% of individuals in many communities, particularly young children, underscoring the need for effective prevention strategies in early life. These allergic conditions have been linked to environmental and lifestyle changes driving the dysfunction of three interdependent biological systems: microbiota, epithelial barrier and immune system. While this is multifactorial, dietary changes are of particular interest in the altered establishment and maturation of the microbiome, including the associated profile of metabolites that modulate immune development and barrier function. Prebiotics are non-digestible food ingredients that beneficially influence the health of the host by 1) acting as a fermentable substrate for some specific commensal host bacteria leading to the release of short-chain fatty acids in the gut intestinal tract influencing many molecular and cellular processes; 2) acting directly on several compartments and specifically on different patterns of cells (epithelial and immune cells). Nutrients with prebiotic properties are therefore of central interest in allergy prevention for their potential to promote a more tolerogenic environment through these multiple pathways. Both observational studies and experimental models lend further credence to this hypothesis. In this review, we describe both the mechanisms and the therapeutic evidence from preclinical and clinical studies exploring the role of prebiotics in allergy prevention.

## 1. Introduction

Allergic diseases (atopic dermatitis (AD), respiratory allergies and food allergies (FA)) are a mounting public health burden—now classified as the 4th most common global disease by the World Health Organization. Affecting up to 30% of the world population [1,2], allergic diseases are also the earliest onset non communicable diseases (NCDs) and a clear manifestation of the vulnerability of the immune system (IS) to modern environmental changes.

The development of allergic diseases has been linked to dysfunction of complex mucosal systems, which collectively comprise of the microbiota, the epithelial barrier and the IS—operating as an interdependent functional unit to maintain both host protection and immune tolerance. In an allergic state, these processes are disrupted, most notably with the failure of immune tolerance, albeit to specific allergen targets. The reasons for this specificity are still unclear, but generally associated with alternations in the T regulatory cell (Treg) and T helper (Th) cell function. The resultant shift in a Th1/Th2 cytokine balance towards a Th2 dominance is associated with the overproduction of interleukine 4 (IL-4), and allergen specific immunoglobuline E (IgE) [3].

There are now numerous studies showing altered microbial signatures in infants who subsequently develop allergic diseases [4,5,6,7,8,9,10], including studies suggesting a critical window during which these alterations may be a specific risk factor for later allergic diseases. This provides a plausible causal pathway for allergy pathogenesis. There are numerous environmental changes that may be contributing to changes in human microbiota profiles. These include dietary changes [11], especially more ultra-processed foods and reduced fiber and fermentable foods consumption, reduced fresh foods and natural microbial loads in our food supply, widespread use of antibiotics and antimicrobial products in healthcare, environmental toxicants, and reduced contact with biodiversity and natural environments. 

Pre-symptomatic changes in the immune function have been detected at birth in children who develop a subsequent allergic disease [1,6,12,13], also suggesting that the impact of environmental change is occurring very early in development—also reflected in the early onset of conditions such as AD and FA, often within the first months of life [14]. Again, this underscores the importance of very early interventions to prevent allergic diseases. Indeed, it is already well-documented that an exposure to a range of environmental factors during pregnancy, delivery and lactation, have the capacity to influence the immune development of the fetus and newborn [15,16]. More recently, this has been shown to include the transfer of bacteria and immune factors from the mother to the fetus or child [6,7] all with the capacity to influence the establishment of both microbiota and the IS. This is the basis for current studies specifically focusing on dietary nutrients known to modulate the microbiome during these critical periods—namely prebiotics. Indeed, both human studies and experimental animal models show that consumption of prebiotics by the mother and her offspring appears to influence the development of both the microbiota and IS. 

The aim of this review is to review these concepts, describe the mechanisms of prebiotic effects on the microbiota, IS and epithelial barrier, and how these may be of value in allergy preventive strategies.

## 2. Prebiotics

### 2.1. First Generation of Prebiotics

Prebiotics were first described by Gibson and Roberfroid in 1995 as “non-digestible food ingredients that beneficially influence the health of the host by stimulating the activity of one or more commensal colon bacteria” [17]. The acquisition of new scientific data on their mode of action and their specificity allowed refining of this definition to being requalified in 2017, by the International Scientific Association of Probiotics and Prebiotics (ISAPP), as “substrates selectively used by micro-organisms of the host conferring benefits for his health”. Prebiotics must meet three criteria: 1: Be resistant to digestion in the stomach and upper intestine, 2: Be fermentable by the gut microbiota and 3: Specifically stimulate the growth and/or activity of intestinal bacteria beneficial to our health [18]. The benefits of prebiotics are not limited to the gut, they can also act systemically [19]. Indeed, new original studies demonstrate that prebiotics can also modulate IS and facilitate many biologic processes, including infection prevention and the improvement of mood and memory [19]. 

Prebiotics are usually composed of linked sugars such as oligosaccharides and short-chain polysaccharides (see Figure 1). These molecules have the chemical characteristic of being indigestible by the enzymes present in the intestinal tract. They will therefore be able to serve as nutritional substrates for microorganisms considered “beneficial” such as *Bifidobacterium* and *Bacteroides* or come into direct contact with the surrounding cells [20]. Fructans such as fructo-oligosaccharides (FOS) and inulin, as well as galactans such as galacto-oligosaccharides (GOS) are the most studied prebiotics for their modulatory effects on the microbiota. The main proven and assumed prebiotics are shown in Table 1 (from Afssa, 2005). The prebiotics that are most commonly found to occur naturally in our foods are the FOS and inulin. They are found in plant-based foods such as in some vegetables (leek, onion, garlic, artichoke, chicory and asparagus), fruits (banana) and cereals (rye, corn). With a balanced European diet, 3–11 g of natural prebiotics can be consumed per day [21]. In comparison, only 1–4 g is usually consumed daily in the USA. Prebiotics are also produced commercially as supplements through the hydrolysis of polysaccharides or enzymatic reactions from lower molecular weight sugars. Although there are many commercially available foods and food ingredients that claim to be prebiotics, currently only lactulose, FOS and GOS have a proven prebiotic effect and status.

### 2.2. Second Generation of Prebiotics

A second generation of prebiotics was designed in the 2000s to improve their functionality. The strategy consisted of adapting the chemical structure of prebiotics by a digestion with a specific enzyme produced by a probiotic. The prebiotic Bimuno (Clasado Biosciences Ltd.) is a good example. It corresponds to a GOS mixture produced from lactose using enzymes from the probiotic *Bifidobacterium bifidum* NCIMB 41171. It has:-a highly selective and powerful prebiotic effect (growth of beneficial *Bifidobacterium* and improved colonization resistance) [23]. It has been demonstrated to be effective as a prebiotic in healthy adults, healthy elderly adults, irritable bowel syndrome (IBS) sufferers and overweight adults [24,25,26], an anti-invasive function (increasing protection against bacterial pathogens). It significantly reduces pathology and colonization associated with food-borne salmonellosis. It also reduces the incidence, severity and duration of traveller’s diarrhea [27,28,29]. -the ability to interact directly with the IS improving barrier function in the gut. Its positive immunomodulatory effect was further demonstrated in overweight adults with metabolic syndrome, where it significantly increased gut immune parameters involved in the protection against pathogens, as well as reducing blood and faecal inflammatory markers [25].

### 2.3. Human Milk Oligossacharides

Prebiotics are present in human milk and they are called human milk oligossacharides (HMOs). Human milk and colostrum are composed of oligosaccharides (5 g/L to 23 g/L) containing a lactose-reducing end elongated with fucosylated and/or sialylated N-acetyllactosamine units. There are over 150 HMOs structures that vary in their size, charge, and sequence [30]. The most frequent HMOs are the neutral fucosylated and non-fucosylated oligosaccharides [31]. The amount and structure of these HMOs can be really different between women and is dependent upon the Secretor and Lewis blood group status (see Figure 2). A deficiency of α 1, 2-linked fucosylated oligosaccharides in human milk is linked to the mutations in the fucosyltransferase 2 (FUT2) secretor gene. HMOs give no direct nutritional value to the infant, and are minor absorbed across the intestinal wall [32]. Instead, it is suggested, that HMOs can play many other roles for the infant. They are preferred substrates for several species of gut bacteria and act as prebiotics, promoting the growth of beneficial intestinal flora and shaping the gut microbiome. Short-chain fatty acids (SCFA) induced by the gut microbiome fermentation of HMOs are critical for intestinal health [33]. They stimulate the growth of gut commensal bacteria along with giving nourishment for epithelial cells lining the gut. HMOs also directly modulate host-epithelial responses, favoring reduced binding of pathogens to the gut barrier. Intestinal microbiota composition varies between formula-fed and breastfed infants, maybe due to the lack of HMOs in infant formula milk [34]. HMOs act probably as decoy receptors, inhibiting the binding of enteric pathogens to prevent infection and subsequent illness. Furthermore, HMOs induce a selective advantage for colonization by favorable bacteria, thereby inhibiting the growth of pathogenic species.

The HMO composition follows a basic blueprint shown in the center. HMOs contain five different monosaccharides in different numbers and linkages, namely glucose (blue circle), galactose (yellow circle), N-acetlylactosamine (blue square), fucose (red triangle), and sialic acid (purple diamond). All HMOs carry lactose at the reducing end. Lactose can be fucosylated or sialylated to induce the small HMOs 2′-fucosyllactose and 3-fucosyllactose or 3′-sialyllactose and 6′-sialyllactose, respectively (upper left corner). Alternatively, lactose can be elongated with type 1 or type 2 disaccharide units to form linear or branched HMOs (upper right corner). Elongated HMOs then can be sialylated (lower left corner) or fucosylated (lower right corner) or both sialylated and fucosylated (not shown). The HMOs in this figure are only a few relatively simple examples. So far, more than 150 different HMO structures have been characterized [35].

### 2.4. Prebiotics Mechanisms

Prebiotics can influence the host health by two distinct mechanisms: indirect (see Figure 3) or direct (see Figure 4). Indirectly, the prebiotics acts as a fermentable substrate for some specific commensal host bacteria. This nutrient source allows the growth of specific taxa and lead to a modulation of gut intestinal microbiota. The SCFAs released in the gut intestinal tract influence many molecular and cellular processes. Recently, new research explored the direct effect of prebiotics on several compartments and specifically on different patterns of cells (epithelial and immune cells).

Indirect Effects: The Short Chain Fatty Acids (see Figure 3)

Prebiotics are highly fermentable food ingredients. This feature promotes expansion and stimulates the implantation of some beneficial and bifidogenic bacteria. Indeed, it was shown that inulin consumption increases specifically and significantly the abundances of *Bifidobacteria* and *Lactobacilli* [17,36,37]. These gut microbiota modifications improve the host health, notably by inhibiting pathogen implantation in the gut. The increase of *Bifidobacteria* following prebiotic consumption was correlated with an increase of acetate production by *Bifidobacteria*, a decrease of the *C. difficile* pathogen population in the gut intestinal tract and an inhibition of pathogens translocation from the gut lumen to the blood [38,39]. The benefits of *Bifidobacteria* are well described in the literature, recent studies reported that new prebiotics (apple pectin and 1-kestose) efficiently stimulate the proliferation of *Faecalibacterium prausnitzii* known to possess an anti-inflammatory effect [40]. Indeed, the pectin promotes the expansion of *Faecalibacterium prausnitzii,* but also *Eubacterium eligens* DSM3376, which strongly improves in vitro the secretion of the anti-inflammatory cytokine IL-10 [41].

Prebiotics, by their ability to be fermented by bacteria, induce the production of SCFAs and thus act indirectly on health. SCFAs can be used in the gut intestinal tract by the microbiota for their own metabolism or released in the lumen. In the lumen, SCFAs can specifically interact with different cells such as intestinal epithelial cells (IEC) or innate/adaptive immune cells to modify various cellular processes as well as gene expression, differentiation, proliferation and apoptosis. SCFAs can activate G protein coupled receptors (GPRs), such as GPR41 or the free fatty acid receptor (Ffar)3 (acetate = propionate > butyrate), GPR43 or Ffar2 (butyrate = propionate > acetate), GPR109a (butyrate) and olfactory receptor (Olfr)-78 (propionate = acetate), to modulate cell development, function and survival [42]. They can also enter directly into the cells through transporters solute carrier family 16 member 1 (Slc) 16a1 and Slc5a8 or by passive diffusion to subsequently induce signaling pathways [43,44]. Signaling pathway induced by SCFAs-binding are mediated by various actors: protein kinases, such as the AMP-activated protein kinase (AMPK) [45], mitogen-activated protein kinases (MAPK) [46], mammalian target of rapamycin (mTOR), signal transducer and activator of transcription 3 (STAT3) [47] or nuclear factor kappa-light-chain-enhancer of activated B cells (NFkB) [48]. Either by downstream signaling pathways or directly, SCFAs modulate the function of several enzymes and transcription factors including histone acetyltransferase (HATs) or histone deacetylases (HDACs) [49]. Consequences of SCFAs-related genes transcription modification is well described: modification of cellular cycle, anti-microbial effects and metabolism (regulation of lipogenesis and lipolysis in the cytosol of hepatocytes and adipocytes, and action on central appetite regulation) [42]. In the next section of this review, we focus on pro- and anti-inflammatory effects induced by SCFAs on epithelial and immune cells. 

#### 2.4.1. Epithelial Cells

The SCFAs interact directly with IEC in the intestinal tract. It was demonstrated that this interaction can influence the intestinal protective immunity via IEC cytokines secretion by Kim et al. who have shown that SCFAs activate GPR41 and GPR43 on IEC, promoting an inflammatory response [50]. Notably, the production of chemokines (C-X-C Motif Chemokine Ligand (CXCL)10, CXCL2, and Chemokine (C-C motif) ligand (CCL)2) and cytokines (IL-12, IL-1) was induced. On the other hand, GPRs-dependent activation of SCFAs is implicated in the regulation of antimicrobial peptide expression on IEC activating mTOR and STAT3 signaling. These mechanisms protect the host by preventing the pathogenesis implantation in the gut [47]. By modulating the activity of IEC, SCFAs can indirectly interact with immune cells such as dendritic cells (DCs). It has been shown that SCFAs increase the vitamin-A converting enzyme in IEC leading to a raised number of intestinal tolerogenic DCs and Treg [51]. SCFAs also promote the gut barrier function itself inducing the production of IL-18, a cytokine that contributes to the intestinal epithelium homeostasis [52,53]. SCFAs can also be found in peripheral blood where they can potentially act on other epithelial cells [54]. Indeed, Qian et al. have shown that feeding asthmatic mice with SCFAs (butyric acid) could reduce the damage of the lung epithelial barrier in asthmatic patients and decrease airway inflammation [55].

#### 2.4.2. Immune Cells

The SCFAs-derived from microbiota fermentation can directly influence the phenotype or/and activity of various innate and adaptive immune cells [42]. Some studies reported the implication of SCFAs in allergy protection. Trompette et al. have shown that asthmatic mice which received a high-fiber diet had a significant increase of blood SCFAs levels and were protected against the allergic airway disease [56]. They demonstrated that circulating SCFAs can go to the bone marrow to alter hematopoiesis characterized by enhanced generation of macrophage and DC precursors and subsequent seeding of the lungs by DCs with high phagocytic capacity but an impaired ability to promote the Th2 cell function. They conclude that SCFAs can shape the immunological environment in the lung and influence the severity of allergic inflammation. Cait et al. also investigated the mechanism of SCFAs to decrease the allergic airway inflammation and demonstrated that DCs in contact with SCFAs are less able to stimulate T cells, to migrate in response to CCL19 in vitro, and to transport inhaled allergens to lung draining nodes [57]. Thorburn et al., demonstrated that SCFAs enter the bloodstream and inhibit HDACs, leading to the transcription of forkhead box P3 (Foxp3) [58]. Foxp3 stimulates the number and the function of Treg, which suppresses the airway inflammation. Additionally in their study, adult offspring (after prebiotics were fed to pregnant mice) were unable to develop the airway disease. These effects of prebiotic supplementation were mediated in utero independently of microbial transfer as SCFAs are able to cross the placenta. Once in the fetus, SCFAs influence gene expression in the fetal lung, such as natriuretic peptide A, this encodes the atrial natriuretic peptide, a molecule implicated in the modulation of epithelial physiology and IS [58].

In the context of FA, Tan et al. have shown that a high-fiber dietin mice promoted oral tolerance and protected from FA via SCFAs signaling [59]. The molecular mechanism highlighted leading to food tolerance was the enhancement of the retinal dehydrogenase activity in CD103^+^DC, promoting differentiation of Treg cells. This specific diet also increased the IgA production and enhanced the T follicular helper and mucosal germinal center responses.

In conclusion, prebiotics can indirectly effect the gut and the IS by SCFA production. By its regulatory effect, SCFAs can potentially influence and prevent different diseases such as allergies. Nevertheless, the effect of SCFAs and modification of microbiota are not the only mechanism of prebiotics.

### 2.5. Direct Effect of Prebiotics (see Figure 4)

### 2.6. IEC

It is well described that prebiotics modulate the gut microbiota leading to the decrease of intestinal inflammation. However, some studies suggest that oligosaccharides may exert an anti-inflammatory effect *per se*, also called “non-prebiotic effects”. When ingested, prebiotics enter the intestine and are in direct contact with the gut epithelial cells. Zenhom et al., have demonstrated in vitro the anti-inflammatory effect of prebiotics on enterocyte lineages, characterized by reduced IL-12 secretion in Caco-2 cells and gene expression of IL-12, p35, IL-8, and tumor necrosis factor α (TNF-α) such as the inhibition of the translocation of NF-ĸB factor to the nucleus [60]. This direct effect was the peroxisome proliferator-activated receptor gamma (PPARγ) and peptidoglycan recognition protein 3 (PGlyRP3) dependent. To assess direct effects of prebiotics, Ortega-González et al. also tested prebiotics in IEC in vitro [61]. Prebiotics promoted the production of growth-related oncogene (GROα), monocyte chemoattractant protein 1 (MCP-1), and macrophage inflammatory protein 2 (MIP2) with an efficacy that was 50%–80% that of lipopolysaccharides (LPS). Interestingly, the response was markedly decreased by Toll like receptor 4 (TLR4) gene knockdown highlighting that prebiotics are TLR4 ligands in IEC. This was confirmed in a study where treatment of epithelial cells with FOS regulated the expression of TLR-modulated genes including IL-10, TNF-α, CXCL-8 and CXCL-1 [62].

Moreover, dietary supplementation with a synbiotic (prebiotics short chain (sc) GOS/ long chain (lc) FOS in combination with probiotic *Bifidobacterium breve M-16V*) increases galectin-9 on the surface of IEC and circulating galectin-9 levels in mice and humans. These observations were correlated with a reduction of acute allergic skin reaction and mast cell degranulation [63]. Galectin-9 was shown to decrease mast cell degranulation and promotes Th1 and Treg responses. This study highlighted that a diet enriched with a synbiotic leads to the prevention of allergic symptoms [63].

To demonstrate the direct effects of prebiotics on the maintenance of epithelial barrier function, a study by Wu et al. investigated the application of prebiotics onto immortalized gut-derived epithelial cell lines and human intestinal organoids in the absence of microbes and in the context of epithelial injury caused by a non-invasive human enteric bacterial pathogen [64]. Prebiotics directly promoted barrier integrity to prevent pathogen-induced barrier disruptions involving the induction of select tight junction proteins through a protein kinase C (PKC) δ-dependent mechanism. Réquilé et al. confirmed the effects of prebiotics on the tight junction gene expression [65]. These results demonstrate a specific and direct host-nutrient interaction and elucidate a novel mechanism whereby prebiotics maintain gut homeostasis to protect the host against challenge by enteric pathogens or allergens.

Another study has highlighted the interactions between epithelial cells and DC regulation by prebiotics [66]. The addition of prebiotic alone in the culture medium of DC did not modify their secretion. However, the addition of the epithelial cell culture supernatant incubated with prebiotics on DC increases the IL-10/IL-12 secretion ratio by DC. This suggests that prebiotics induce tolerogenic DCs mediated by molecules secreted by epithelial cells. The interaction of prebiotics with epithelial cells and DCs led to changes in the polarization of T cells. Interestingly, the different fibers tested had differential effects on T-cell polarization. GOS, chicory inulin, wheat arabinoxylan, and barley -glucan increased the production of the Th1 cytokine IFN-γ, whereas the Th1 cytokine TNF-α was decreased by GOS and wheat arabinoxylan. Barley-glucan increased the Th1 cytokine IL-2 and suppressed Th2 responses. GOS was the only prebiotic able to promote functional Treg (increase of IL10 secretion). They conclude that direct modulation of the IECs-DCs crosstalk can induce a regulatory immune phenotype and selecting dietary fibers could be essential for future clinical trials on efficacy in allergy management.

In summary, these studies demonstrate that prebiotics modulate host cell signaling to promote epithelial barrier integrity by direct effects on the intestinal mucosa.

### 2.7. Skin Epithelial Cells

It has been shown in mice, that GOS supplementation prevented trans epidermal water loss and UV-induced erythema via dermal expression of cell adhesion and matrix formation markers CD44, metallopeptidase inhibitor 1 (TIMP)-1, and collagen type 1(Col1), thereby improving the skin’s barrier properties [67]. It has also been shown in an AD mouse model that skin inflammatory cell infiltrations (such as skin Th2-related cytokines, TSLP and IL-4) were significantly reduced by treatments with FOS [68]. Moreover, CD4+ Foxp3+ Treg cells were significantly increased in skin lymph nodes. Finally, also in a mouse model of AD, supplementation with prebiotics (Konjac glucomannan) inhibits a scratching behavior and skin inflammatory immune responses by preventing germline class-switching and IgE production [69]. Prebiotic dietary supplementation significantly suppressed eczematous skin lesions, dermal mastocytosis and eosinophilia. Concomitantly, cutaneous overproductions of substance P, IL-10, IL-4, and TNF-α were all inhibited [70].

The prebiotic effects on skin epithelial cells in humans were confirmed in a double-blind, randomized trial on adult healthy subjects divided into two groups, control and GOS oral supplementation for 12 weeks [71]. The authors reported that GOS supplementation was beneficial to the skin characterized by better corneometer values and reduced transepidermal water loss (TEWL). In addition, the differences in total and percentage of wrinkle areas between the two groups were statistically significant after 12 weeks of GOS treatment. Finally, GOS can also prevent keratin depletion caused by phenolic compounds. In another trial involving patients with skin damage due to diabetes, 4 weeks of daily application of emollient containing prebiotics improved biophysical parameters of the epidermis [72]. They observed a decrease of the TEWL and skin dryness, and the restoration of skin sebum levels. Normalization of skin pH was suggested as the beneficial mechanism through improved epidermal skin integrity and limitation of bacterial infection.

In conclusion, both oral supplementation and cutaneous application with prebiotics have an impact on the biophysical parameters of the epidermis, improving the skin’s barrier properties. Knowing that the skin is an important route for allergen sensitization in AD infants [73], it would be interesting to study the effect of prebiotics application on the skin in high-atopy-risk infants to potentially reduce the risk of allergen skin sensitization.

### 2.8. Lung Epithelial Cells

In asthmatic airways, repeated epithelial damage and repair occur and the disrupted epithelial barrier and epithelial dysfunction are crucial in the induction and maintenance of asthma symptoms. No current therapy directly targets this process. Michael et al. have shown that treatment with prebiotics (Mannan Derived from Saccharomyces cerevisiae) stimulated cell spreading and facilitates wound repair in human bronchial epithelium, involving mannose receptors [74]. Prebiotics also increased the expression and activation of Krüppel-like factors (KLFs) 4 and 5, key transcription factors for epithelial cell differentiation, survival, and proliferation.

### 2.9. Immune System

The intestinal IS, also called gut-associated lymphoid tissue (GALT), is a secondary lymphoid organ involved in the processing of antigens that interact with the gut mucosa and the diffusion of the immune response. Prebiotics can be absorbed through the intestinal barrier and may thus be in direct contact with circulating IS cells. Inulin and FOS induce the secretion of IL-10, IL-1β and TNF-α by blood monocytes [75]. This secretion is due to the activation of the NF-κB pathway by the binding of the TLR4. In contrast, FOS and inulin do not have a significant impact on cytokine secretion by T lymphocytes. Possible direct effects of prebiotics are thought to entail ligation of pathogen recognition receptors (PRRs) on the surface of intestine DCs [76]. These PRRs involved in prebiotics signaling include TLRs, C-type lectin receptors (CLRs), NOD-like receptors (NLRs), and galectins. On DC derived from human blood monocytes, GOS and FOS induce IL-10 secretion stimulated by TLR4 binding [77]. Increased secretion of IL-10 by DC leads to the induction of regulatory Foxp3+ T cells. However, Perdijk et al. have shown that HMOs (6′-SL and 2′-FL) and GOS do not alter DC differentiation or maturation of in vitro differentiated DC types [78]. It was also shown that prebiotics supplementation in rats acted at the level of GALT enhancing the production of IL-10 and IFN-γ by CD4+ T lymphocytes in Peyer’s patches as well as the production of IgA in the caecum, compared with controls [79]. Caecal IgA secretion was shown to be dependent on the degree of polymerization of prebiotics [80]. Another study on mice also showed that FOS supplementation increased the secretion of IFN-γ and IL-10 in CD4+ T cells derived from Peyer’s patches [81].

## 3. Clinical and Preclinical Studies

Considering the regulatory effects of prebiotics on epithelial barriers, the IS and the microbiota, several research teams have investigated prebiotic supplementation as a new strategy to prevent allergies in animal models (see Table 2) and in human trials (see Table 3). The prebiotics were given at different times of life, including during pregnancy or lactation. Supplementation was followed by assessing allergic disease development and characterization of the IS, microbiota and physiological parameters.

### 3.1. Preclinical Studies (Table 2)

### 3.2. Prebiotic Supplementation in Adult Mice

To prevent antigen-specific skin inflammation, the strategy used by Watanabe et al. was to examine whether FOS supplementation could favorably change the population of intestinal microbiota to promote immune health [82]. They observed that the number of intestinal *Bifidobacterium* increased in correlation with a decrease in the severity of the skin allergy. Moreover, supplementation of allergic mice with scGOS/lcFOS and *Bifidobacterium breve M-16V* protected against acute allergic skin reactions [63]. This was correlated with Galectin-9 expression in the gut and in the sera, and reduced mast cell degranulation. *In vitro*, galectin-9/scGOS/lcFOS /*Bifidobacterium breve M-16V* exposure enhanced Th1 cell differentiation in mesenteric lymph nodes cells of mice and in peripheral blood mononuclear cells.

Several animal model studies have demonstrated that prebiotic supplementation could be interesting to protect against FA. Consumption of FOS significantly decreased the frequency of mast cells and the rate of edematous development in the duodenum when FA was introduced in mice [83], such as attenuation of intestinal Th2 cytokine responses [84]. In addition, supplementation with fibers (guar gum and cellulose (35% crude fiber)) had a protective effect on peanut allergy in mice *via* the modification of the gut microbiota (increase of *Bacteroïdes*, *Lactobacillus* and *Bifidobacterium* and decrease of *Firmicutes*) and increased SCFAs secretion [59]. The suppression of the allergic responses were followed by 1) enhanced retinal dehydrogenase activity in CD103+ tolerogenic DCs, 2) increased IgA production associated with less specific IgE, 3) enhanced T follicular helper activation and IFN-γ secretion, 4) decreased eosinophil infiltration and goblet cell metaplasia inhibited serum specific IgE levels, but, increased IFN-γ [59]. Interestingly, Schouten and al. have demonstrated that cow milk allergy symptoms are reduced in mice fed with synbiotics both during oral sensitization with whey [85] and post-sensitization [86]. The dietary intake of FOS during the development of FA was also able to attenuate the production of gut Th2 cytokine responses by regulating early activation of naive CD4+ T cells, which induce both Th1 and Th2 cytokines. These data suggested FOS might be a potential food agent for FA prevention by modulating oral sensitization to food antigens [84].

In the field of allergic asthma, dietary supplementation with FOS attenuated airway inflammation in mice by suppressing the expression of IL-5 and eotaxin in the lungs [87,88]. Decreasing concentrations of allergen specific IgG1, Th2-related cytokines and chemokines were also observed such as a reduced eosinophilia. Vos et al. confirmed that a mixture of GOS/FOS decreased allergic asthma by inhibiting the Th2 response and by inducing the activation of the Th1 response [89].

To reinforce the allergy protective effect, some studies have combined the prebiotic supplementation in offspring just after weaning and in the mother. This strategy promoted the maintenance of tolerance pathways in mice food allergic to wheat in contrary to the post weaning administration alone [90]. Prebiotics reinforced gut barrier functions and induces higher tolerance and expression of associated biomarkers (IgA, tumor growth factor b (TGF-β) in milk, IgG2a and IgA) [91,92]. Fujiwara et al. showed that the consumption of FOS by the mother and pups decreases the severity of AD in the offspring [92]. The symptoms of AD and the rate of IgG1 and TNF-α were lower in FOS supplemented group. These interesting results have encouraged researchers to evaluate the allergic preventive effect of maternal prebiotic supplementation during pregnancy and lactation. 

### 3.3. Offspring Prevention by Maternal Intervention

Another strategy tested in mice models was the prevention of allergies by prebiotic supplementation to mothers during pregnancy alone or combined to lactation. Supplementation during pregnancy alone caused a reduction of airway allergic symptoms in the offspring [93]. The descendants showed a higher level of Treg than the control group, but the rate of IgE, IgG1, IgG2a and the immune cell count in the bronchoalvolar lavage were similar. Sensitized mothers exposed to prebiotics during pregnancy were more able to protect the offspring against cutaneous allergy than control sensitized mothers [94]. Interestingly, prebiotic exposure during lactation of non-sensitized or sensitized mothers leads to a significant decrease in allergic symptoms in the offspring, showing that the mother’s sensitization state does not influence the effect of prebiotics when given during the lactation.

In the context of FA, exposure to GOS/inulin prebiotics during gestation and lactation resulted in a reduced allergic response to wheat in the offspring [95]. This protective effect was measured by a reduction of clinical symptoms (clinical score, IgE, histamine release), was associated with a stronger intestinal permeability and a modification of microbiota in mothers and their offspring. They observed an enhancement of the regulatory response to allergic inflammation and changes in the Th2/Th1 balance toward a dampened Th2 response in mice from GOS/inulin mixture-supplemented mothers. Prebiotics reinforced the gut barrier function in the offspring with a higher expression of mucine 2, a decrease of paracellular and transcellular permeability and a decrease in CD23 expression [95].

In an AD mouse model, Fujiwara et al. showed that the consumption of FOS by the mother during gestation and lactation decreases the severity of AD in the offspring (decrease of skin severity score and scratching frequency) [92]. Some markers of inflammation (TNF-α) and Th2 cells response (IgG1) were decreased. 

These results obtained in animal models suggest that the modulation of gut microbiota, epithelial barrier and IS by prebiotics may be beneficial for allergy prevention. This protective effect observed after a maternal and pup supplementation suggests that pregnancy and breastfeeding are crucial periods during which biological systems can be significantly modulated to protect against future allergic diseases in the offspring.

### 3.4. Human Clinical Trials

In parallel to these preclinical studies, several human clinical trials using prebiotics as an allergy prevention strategy were undertaken (summarized in Table 3), but only by intervening in infancy by supplementation of infant formula. In order to give the recommendations in the World Allergy Organization (WAO) clinical practice guidelines for allergies prevention or Guideline for Allergic Disease Prevention (GLADP), Cuello-Garcia et al. conducted a systematic review including 22 randomized trials with a minimum follow-up of four weeks, comparing any type of prebiotic against placebo or no prebiotic [96]. The following outcomes assessed in these trials in children were the induction of: AD, asthma and/or recurrent wheezing, FA, and allergic rhinitis. Six studies (2030 participants) assessed the development of AD [96,97,98,99,100,101,102,103,104] and found, with regard to the risk of AD development, the relative risk was 0.68; 95%CI, 0.40 to 1.15 when compared to the control group. In the two studies reporting the induction of wheezing/asthma [97,99], prebiotics were able to decrease the risk of asthma or wheezing when compared to the control group (RR, 0.37; 95% CI, 0.17 to 0.80; 249 participants). However, precision was a concern due to a low number of events and to what extent the definition of “recurrent wheezing” reflected a true reduction in the risk of developing asthma. Moreover, no effect of prebiotics on recurrent wheezing was found at five years of age [97]. Only Ivakhnenko et al. investigated the development of FA [99]. In this trial, prebiotics were able to reduce the risk of FA when compared to placebo (RR, 0.28; 95% CI 0.08 to 1; 115 participants). Finally, Sierra et al. reported a composite of the induction of any allergy with no difference between the groups (RR, 1.39; 95% CI 0.91 to 2.12; 264 participants) [105]. However, the number of events was small and the confidence interval did not exclude a possible large increase in the development of any allergies. Cuello-Garcia et al. concluded that the currently available evidence on prebiotic supplementation in infancy to reduce the risk of developing allergies is very uncertain [96].

Since this meta-analysis, two further clinical trials have been conducted. Wopereis et al. demonstrated that early-life nutrition with prebiotics (0.8 g/100 mL GOS: FOS: pAOS from birth to 26 months of age in infants at risk) had an impact on the establishment of the infant gut microbiota with a potential link between the activity of the microbiota and the expression of eczema in early life [106]. Ranucci et al. recently published the results of their clinical trial based on a 48 weeks supplementation with GOS/polydextrose (PDX) of high-atopy-risk infants due to parental history of allergic diseases [107]. The cumulative number of infants with AD was not statistically different between infants supplemented or not with GOS/PDX at 36 weeks, 48 weeks and 96 weeks of age. The number of respiratory infection episodes until 48 weeks were lower in infants supplemented with FOS/GOS than for infants fed standard formula (*p* = 0.023) but no longer detected at 96 weeks. In the same way as for the previous studies, *Bifidobacteria* and *Clostridium* cluster I colonization increased over time in the supplemented group.

To summarize, there is currently not enough robust evidence from human infant trials to support a beneficial effect of supplementation with prebiotics for allergy prevention.

So far, the only specific recommendations for the use of prebiotics for allergy prevention have come from the WAO, 2016. An expert panel concluded that, there is a likely net benefit of prebiotics [108]. The WAO guideline panel suggested: 1: prebiotic supplementation in not-exclusively breastfed infants, both at high and at low risk for developing allergy (conditional recommendation, low certainty of evidence); 2: that clinicians and parents do not use prebiotic supplementation in exclusively breastfed infants (conditional recommendation, low certainty of the evidence).

No human studies in which prebiotics have been given to mothers during pregnancy and/or breastfeeding have been published to date, however two clinical studies are currently being conducted: SYMBA (Trial Identification Number: ACTRN12615001075572) and PREGRALL (Trial Identification Number: NCT03183440). SYMBA is a double-blinded, randomized controlled trial investigating the effects of maternal prebiotic supplementation (GOS:FOS in a ratio 9:1) from 18–20 weeks gestation during pregnancy until six months of lactation on the development of infant allergic disease [109]. PREGRALL is a randomized, multi-center, double-blinded, trial which aims to evaluate the effectiveness of gestational prebiotic supplementation (from 20th week of gestation until delivery, 11.8 g of 9:1 GOS/inulin powder per day) versus placebo on the occurrence of AD at one year of age in high risk children (defined as having a maternal history of atopic disease) [110].

## 4. Conclusions

Despite intensive research, there is still no cure nor totally effective prevention strategies for allergic diseases. However, future strategies based on prebiotics supplementation to prevent allergies may be promising. Indeed, prebiotics can directly or indirectly modulate the three major systems that are dysfunctional in allergic disease development: the microbiota, the IS and the epithelial barriers. Preclinical models have proven the efficacy of prebiotic supplementation to overcome skin, food and respiratory allergic symptoms. However, infant supplementation with prebiotics has showed limited effects. Maternal prebiotic supplementation during pregnancy to prevent allergy in children seems to be the best window of opportunity to modulate biological systems. Two clinical studies are ongoing in this field to decipher if pregnant women could benefit from prebiotic supplementation to reduce the risk of allergies in their children.

## Figures and Tables

**Figure 1 nutrients-11-01841-f001:**
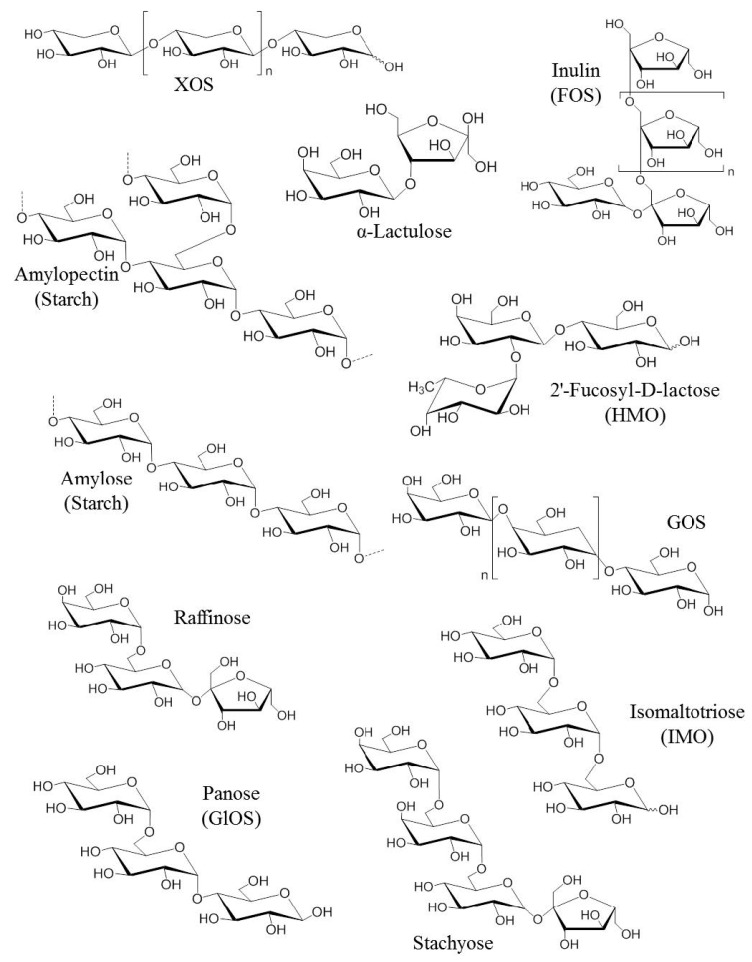
Chemical structures of the first generation of prebiotics. FOS: Fructo-oligosaccharides, GlOS: Gluco-oligosaccharides, GOS: Galacto-oligosaccharides, HMO: Human milk oligosaccharides, IMO: Isomalto-oligosaccharids [22].

**Figure 2 nutrients-11-01841-f002:**
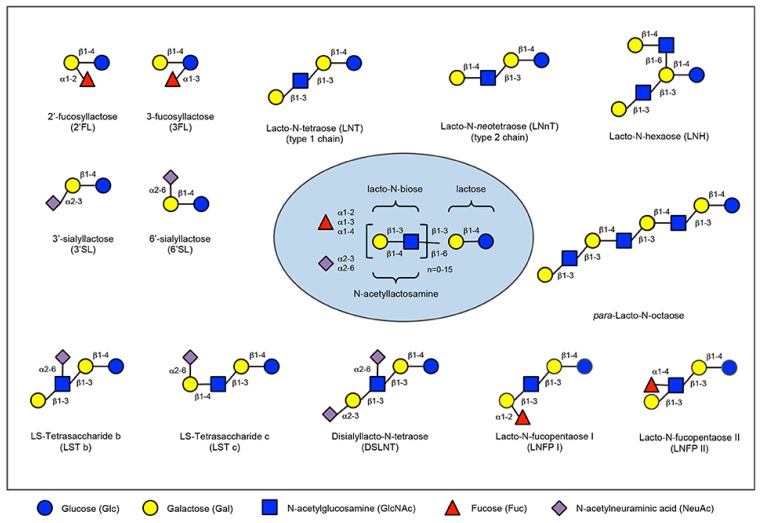
Human milk oligosaccharide composition blueprint.

**Figure 3 nutrients-11-01841-f003:**
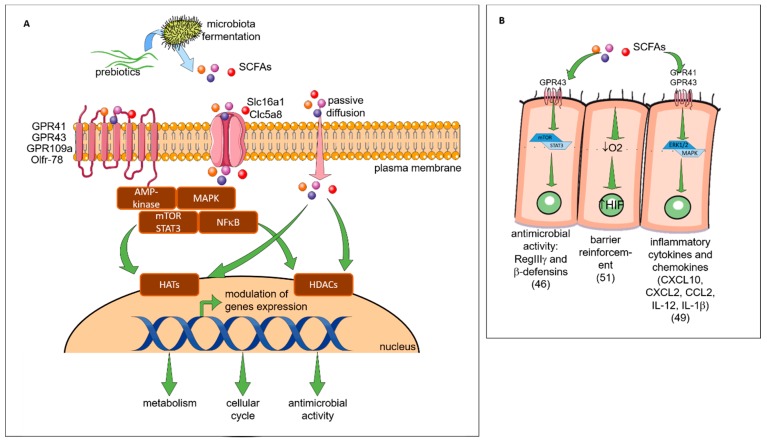
Indirect effects of prebiotics. (**A**) The general mechanisms of SCFAs. The SCFAs are metabolites derived from the fermentation of prebiotics by the microbiota. They are consumed by the microbiota or released into the biological systems (blood, gut, lung, placenta). They can interact with the cells by three mechanisms. In the first mechanism, the GPRs which are receptors coupled to signaling pathways (AMP-activated protein kinase (AMP-K), mammalian target of rapamycin (mTOR), signal transducer and activator of transcription 3 (STAT3), mitogen-activated protein kinases (MAPKs), nuclear factor kappa-light-chain-enhancer of activated B cells (NF-κB)) are involved. The second mechanisms correspond to the diffusion channels (solute carrier family 16 member 1 (Slc) 16a1 and 5a8) that allow the SCFAs transport directly to cytoplasm and their potential interactions with pathways. By these two mechanisms, the signaling cascade is activated and can influence the transcription of genes by acetylation and deacetylation respectively via the histone acetyltransferases (HAT) and the histone deacetylases (HDAC) enzymes (epigenetic mechanisms). In the last mechanisms, there is a passive diffusion of SCFAs able to modulate directly enzymes (HDAC, HAT) involved in epigenetic processes. The modulation of genes expression by acetylation and deacetylation will have different consequences such as a modification of metabolism, cell cycle or microbial activity described in Figure 3B,C. (**B**) The specific impact of SCFAs on epithelial cells. SCFAs (butyrate, propionate) can interact with the G-protein coupled receptor (GPR)43 receptor and activates the mTOR/STAT3 pathway allowing the modulation of genes to increase the expression of antimicrobial peptides such as regenerating islet-derived protein 3 gamma (RegIIIγ) and β-defensins. SCFAs can directly increase the epithelial barrier function by stimulating O2 metabolism in intestinal epithelial cell lines. This mechanism results in the stabilization of the transcription factor hypoxia-inducible factor (HIF-1). SCFAs interact also with the GPR 41 (acetate, propionate) and GPR43 receptors to activate the extracellular signal-regulated kinases (ERK) 1/2 and MAPK signaling pathway. In this way, epithelial cells produce inflammatory chemokines and cytokines during the immune response to protect the organism against aggressions or infections. The consumption of SCFA also increases the secretion of the antimicrobial peptide by epithelial cells. (**C**) The specific impact of SCFAs on IS. SCFAs can be found in the bloodstream. They can interact with different immune cell subtypes. In a first step they can modify the hematopoeisis of dendritic cells (DC) precursors in the bone marrow and induce CD11c+ CD11b+ DCs in lung-draining lymph nodes. The cells CD11c+ CD11b+ DC have a lower capacity to activate the Th2 cell which results in the reduction of allergic asthma. SCFAs are also able to modify in vitro the functionality of FMS-like tyrosine kinase 3 ligand (Flt3L)-elicited splenic DCs: a lower ability to activate T cell and to transport antigen to the lymph node and a lower expression of chemokine (C-C motif) ligand 19 (CCL19) on their surface decreasing their ability to move in different sites. In the lungs SCFAs are able to inhibit the enzyme HDAC9 resulting in an increase of the forkhead box P3 (FoxP3) transcription factor and then an increase of the number and the activity of Treg. In the intestine the increase of retinoic acid-synthesizing (RALDH) enzyme activity during the consumption of SCFA, allows the conversion of vitamin A to the retinoic acid in tolerogenic DC CD103 +. Then, the retinoic acid acts directly on T cells and induces their differentiation into Treg.

**Figure 4 nutrients-11-01841-f004:**
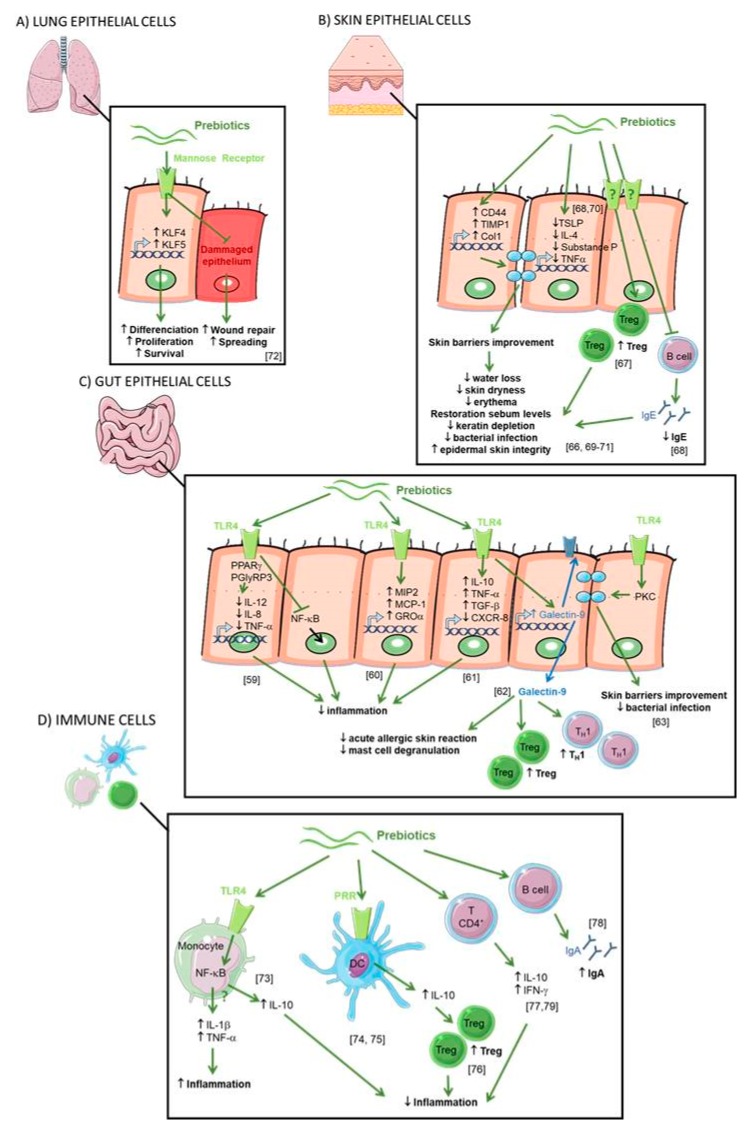
Direct effects of prebiotics. (**A**) Direct effect of prebiotics on lung epithelial cells. Mannan prebiotic stimulates cell spreading and facilitates wound repair in damaged human bronchial epithelium, involving mannose receptors. Prebiotics also increases expression and activation of Krüppel-like factors (KLFs) inducing cell differentiation, survival, and proliferation. (**B**) Direct effect of prebiotics on skin epithelial cells. Prebiotics supplementation improved water retention and prevented erythema via the expression of CD44, metallopeptidase inhibitor 1 (TIMP)-1, and collagen type 1(Col1) improving the skin’s barrier properties. Prebiotics suppress overproductions of thymic stromal lymphopoietin (TSLP), substance P, IL-10, IL-4, and tumor necrosis factor (TNF)-a leading to reduced transepidermal water loss and skin dryness, prevention of keratin depletion, improvement of biophysical parameters of the epidermis, restoration of skin sebum levels, and limitation of bacterial infection. Prebiotics increase CD4+ Foxp3+ Treg in skin lymph nodes and prevent germline class-switching and IgE production. (**C**) Direct effect of prebiotics on gut epithelial cells. Prebiotics are Toll like receptor 4 (TLR4) ligands in IEC. Prebiotics induce a range of anti-inflammatory cytokines and reduce pro-inflammatory cytokines to inhibit gut inflammation. Prebiotics enhance galectin-9 expression correlated with reduced acute allergic skin reaction and mast cell degranulation and promoted Th1 and Treg responses. Prebiotics directly promoted barrier integrity to prevent pathogen-induced barrier disruptions involving the induction of protein kinase C (PKC). **(D)** Direct effect of prebiotics on immune cells. Prebiotics induce both the secretion of anti-inflammatory (IL-10) and pro-inflammatory (IL-1β and TNF-α) cytokines by blood monocytes due to the activation of the NF-ĸB pathway by the binding of TLR4. Prebiotics bind pathogen recognition receptor (PRRs) on the surface of DCs inducing IL-10 secretion and Treg cells. Prebiotics enhance the secretion of IL-10 and interferon-γ (IFN-γ) by CD4+ T cells and IgA.

**Table 1 nutrients-11-01841-t001:** List of proven and assumed prebiotics.

Substance	Composition	Degree of polymerization	Process of obtaining
Fructans	Glucose, fructose		
•Linear			
*Inulin*	β-2,1 bonds	10 to 60	Extraction
*Fructo-oligosides (FOS) and oligofructose (OF)*	β-2,1 bonds	2 to 9	Synthesis, hydrolysis
Levans	β-2,6 bonds	20–30 (from vegetal)*	Enzymatic
•Connected (graminans)	β-2,6 & β-2,1 bonds	unknown	Enzymatic biosynthesis
Lactulose	Galactose, fructose, β-1,4 bonds	2	Chimic synthesis
(Trans)galacto-oligosides (TOS)	Glucose, galactose, β-1,6 bonds	2 to 5	Enzymatic biosynthesis
*Galacto-oligosides (GOS)*	Glucose, galactose, β-1,6 bonds	2 to 5	Enzymatic biosynthesis
Xylo-oligosides (XOS)	Xylose, β-1,4 bonds	2 to 9	Enzymatic hydrolysis
Soy oligosides or α-galactosides (raffinose, stachyose & verbascose)	Galactose, fructose, glucose, β-1,6 and β-1,2 bonds	3 to 5	Extraction
Isomaltooligosides	Glucose, α-1,6 bonds	2 to 5	Enzymatic hydrolysis Enzymatic bioconversion
Oligolaminarans (β-glucans)	Glucose (± mannitol), β-1,3 and 1,6 bonds	5 to 25	Enzymatic hydrolysis
Polydextrose	Poly-D-glucose (glucose 89%, sorbitol 10% and phosphoric acid 0.1%)	12 (mean of DP)	Chimic synthesis
D-tagatose	Tagatose	1	Extraction
Starch resistant	Glucose, α-1,4 and 1,6 bonds	> 1000	Extraction

In italic: *proven prebiotics*. * Levans produced by microorganisms have usually molecular weights superior to 10^6^.

**Table 2 nutrients-11-01841-t002:** Preclinical studies with prebiotics for allergy prevention.

Study design	Ref	Model	Age at the beginning of the protocol	Supplementation period	Diet intervention	Primary outcome
prebiotic supplementation in adult mice	[82]	contact hypersensitivity	5 weeks old mice	during all the protocol	5% lcFOS	Contact hypersensitivity is reduced by supplementation. The number of intestinal *Bifidobacteria* was increased, and *B. pseudolongum* was most represented.
	[63]	skin allergy	6 weeks old mice	during all the protocol	scGOS + lcFOS and *Bifidobacterium breve M-16V* (GF/Bb) diet supplemented with Bb (2% wt:wt, 2 3 109 CFU/g) and scGOS /lcFOS (2%)	The number of intestinal *Bifidobacterium* increased in correlation with a decrease in the severity of the skin allergy. Moreover, supplementation of allergic mouse protected acute allergic skin reaction.
	[83]	FA	6 weeks old mice	during all the protocol	5% FOS	Decreased CCR4-positive cells and mast cells at the site of induction of allergy at the onset of allergy.
	[84]	FA	8–12 weeks old mice	during all the protocol	7,5% FOS	Attenuation of the induction of intestinal Th2 cytokine responses by regulating early activation of naive CD4+ T cells, which produce both Th1 and Th2 cytokines.
	[59]	FA	6–8 weeks old mice	2 weeks prior and throughout experiments	high-fiber diet is enriched inguar gumand cellulose (35%crudefiber). Sodium acetate, propionate, or butyrate was administered in drinking water	Dietary fiber with vitamin A increases the potency of tolerogenic CD103+ DCs and high-fiber diet protects against allergy via gut microbiota Dietary fiber promotes TFH and IgA responses
	[85]	FA	3 weeks old mice	2 weeks prior and throughout experiments	2% of mixture of scGOS/scFOS/lcFOS (9:1)	Symptoms are reduced in mice fed with synbiotics both during oral sensitization with whey
	[86]	FA	4 weeks old mice	during 3 weeks after sensitization	1% of scFOS and lcFOS (9:1)	Cow milk allergy symptoms are reduced in mice fed with synbiotics both during oral sensitization with whey post-sensitization
	[87]	asthma	7 weeks old mice	1 weeks prior and throughout experiments	2,5% FOS	FOS attenuated airway inflammation in mice by suppressing the expression of IL-5 and eotaxin in the lungs
	[88]	asthma	4 weeks old mice	2 weeks before sensitization	1% GOS	Prevention of induction of eosinophilia of the respiratory tract and secretion of Th2-related cytokines and chemokines in the lungs.
	[89]	asthma	5–8 weeks old	during all the protocol	5% scGOS:lcFOS (9:1) + pAOS	Dietary supplementation causes an orientation of the immune response to the Th1 response.
	[90]	FA	6–12 weeks old	gestation + lactation, and until the end of the protocol	4% GOS:Inuline (9:1)	Food supplementation stimulates immune tolerance and strengthens the intestinal barrier.
	[91]	FA	6–12 weeks old	gestation + lactation, and until the end of the protocol	4% GOS:Inuline (9:1)	Increased biomarkers of tolerance after sensitization.
offspring prevention by maternal intervention	[92]	cutaneous inflammation	8 weeks old	gestation + lactation, and until the end of the protocol	5% FOS	The consumption of prebiotics decreases the severity of atopic dermatitis.
	[93]	asthma	8 weeks old	gestation	50% scGOS:lcFOS (9:1)	Decrease of airway hyper-reactivity, specific IgE, increase of specific IgG2a and regulatory T at peripheral level.
	[94]	asthma	10 weeks old	gestation + lactation	5% scGOS:lcFOS (9:1)	Decreased allergic symptoms in both cases
	[95]	FA	8 weeks old	Gestation + lactation	4% GOS:Inuline (9:1)	Modification of the microbiota, protection of the intestinal barrier and induction of immune tolerance.

(lcFOS: long chain fructo-oligosaccharides; scGOS: short chain galacto-oligosaccharides; FA: food allergy; FOS: fructo-oligosaccharides; GOS: galacto-oligosaccharide).

**Table 3 nutrients-11-01841-t003:** Prebiotic supplementation of infant formula trials with allergic disease outcomes.

Study design	Ref.	Outcomes	Population	Intervention	Control	Primary outcome
Randomized, controlled, double-blind trial with a 6- month intervention period from birth to 6 months of age.	[103]	AD at 6 months.	Term infants at risk of atopy due to parental history of allergic diseases.	*n* = 102 EHF formula with added GOS: FOS (0.8 g/100 mL).	*n* = 104 EHF formula with added maltodextrine (0.8 g/100 mL) as placebo.	- At 6 months of age: fewer infants had AD in GOS: FOS group (9.8%) than the control group (23.1%; *p* = 0.014).
	[97]	Follow-up until 2 years of age. AD, recurrent wheezing, number of infectious episodes and allergic urticaria at 2 years of age.	Term infants at risk of atopy due to parental history of allergic diseases.	*n* = 66 hypoallergenic EHF with added GOS: FOS (0.8 g/100 mL).	*n* = 68 hypoallergenic EHF with added maltodextrine (0.8 g/100 mL) as placebo.	- The cumulative incidences of AD, recurrent wheezing, and allergic urticaria were lower in the GOS/FOS group (13.6, 7.6, 1.5%, respectively) than in the placebo group (27.9, 20.6, and 10.3%, respectively; *p* < 0.05). The number of overall infections and the number of fever episodes was lower in the GOS/FOS group than in the placebo group (*p* = 0.01 and *p* < 0.0001 respectively).
	[102]	Follow-up until 5 years of age. Cumulative incidence of allergic manifestations (AD, recurrent wheezing, allergic rhinoconjunctivitis and urticaria) during 5 years, and the prevalence of allergic and persistent allergic manifestations at 5 years.	Term infants at risk of atopy due to parental history of allergic diseases.	*n* = 42 hypoallergenic EHF with added GOS: FOS (0.8 g/100 Ll).	*n* = 50 hypoallergenic EHF with added maltodextrine (0.8 g/100 mL) as placebo.	The 5 years cumulative incidences of any allergic manifestation and DA were significantly lower in the GOS/FOS group (30.9, 19.1 %, respectively) compared to placebo group (66, 38 %, respectively) (*p* < 0.01 and < 0.05). Children in the GOS/FOS group tended to have a lower incidence of allergic rhinoconjunctivitis, and allergic urticaria (4.8 vs 16% for both manifestations, *p* = 0.08). There was no difference in the cumulative incidence of recurrent wheezing. With regard to the prevalences at 5 years, intervention group had significantly lower prevalence of any persistent allergic manifestation and rhinoconjunctivitis (4.8, 2.4 %, respectively) compared to placebo (26, 14 %, respectively) (*p* < 0.01 and = 0.05). Prevalence of persistent AD tended to be lower in the intervention group (2.4 vs 12%, *p* = 0.09). Although intervention group had 75% reduction in the prevalence of persistent wheezing (4.8 vs 14 %), no significance was shown.
	[104]	Levels of total immunoglobulins and of cow’s milk protein at 6 months of age.	Term infants at risk of atopy due to parental history of allergic disease.	*n* = 41 hypoallergenic EHF with added GOS: FOS (0.8 g/100 mL).	*n* = 43 hypoallergenic EHF with added maltodextrine (0.8 g/100 mL) as placebo.	Total IgE, IgG1, IgG2 and IgG3 levels were significantly declined after GOS/FOS supplementation, whereas no significant differences in the levels of total IgG4 were observed between the two treatment groups. Ig levels were similar between infants with our without AD. Cow’s milk proteins-specific IgE levels were very low in all infants but cow’s milk proteins-specific-IgG1 was significantly decreased in the GOS/FOS-supplemented infants (*p* < 0.005).
Randomized, controlled, double-blind trial with a 12- month intervention period from birth to 12 months of age.	[98]	AD at 12 months of age.	Term infants at low atopy risk due to no family history of allergic diseases.	*n* = 414 CM formula with added GOS: FOS (0.8 g/100 mL) and pAOS (0.12 g/100 mL).	*n* = 416 CM formula.	At 12 months of age, fewer infants had AD in GOS: FOS: pAOS group (5.7%) than the control group (9.7%; *p* = 0.04) and AD was less severe among the infants affected in the supplemented group (*n* = 8; median SCORAD score, 8; range, 3–25) than in the control group (*n* = 16; median SCORAD score, 12; range, 2–59; *p* = 0.08).
Randomized, controlled, double-blind trial with a 2- month intervention period from birth to 2 months of age.	[99]	Incidences of AD, obstructive bronchitis, recurrent wheezing, gastrointestinal and upper respiratory tract infections at 18 months of age. Gut microbiota composition at 2 months of age.	Term infants at low atopy risk due to no family history of allergic diseases.	*n* = 80 standard formula with added GOS: FOS (0.8 g/100 mL).	*n* = 80 standard formula	In infants fed with formula supplemented with GOS/FOS, fecal concentrations of *Bifidobacteria* and *lactobacilli* were significantly higher than infants fed with the standard formula (8.92 ± 1.011 and 7.22 ± 0.74 CFU/g for supplemented children respectively vs 7.81 ± 0.83 and 6.81 ± 0.93 CFU/g for non-supplemented children respectively *p* < 0.05). Infants fed with supplemented formula had significantly less allergic reactions to food products compared to the control infants (4.84% vs. 16.98%, *p* < 0,05).The incidence of AD was also the highest in control group (16.98% vs. 4.84% accordingly; *p* < 0.05).
Randomized, controlled, double-blind trial with a 6- month intervention period from birth to 6 months of age.	[101]	AD at 12 months of age.	Term infants at risk of atopy due to parental history of allergic diseases.	*n* = 375 PHF formula with added GOS:FOS:pAOS (0.8 g/100 mL).	*n* = 383 CM formula.	At 12 months of age: There was no difference in the incidence of AD between the PHF GOS: FOS: pAOS intervention group (28.7%) and the CM formula control group (28.7%; *p* = 0.90).
Randomized, controlled, double-blind trial with a 26- month intervention period from birth to 26 months of age.	[106]	Fecal pH, bacterial taxonomic compositions and microbial metabolite levels in the first 26 weeks of life. To identify microbial patterns associated with the onset of eczema.	Term infants at risk of atopy due to parental history of allergic diseases.	*n* = 51 PHF formula with added GOS: FOS (0.68 g/100 mL) and pAOS (0.12 g/100 mL).	*n* = 57 standard formula	Fecal microbial composition, metabolites, and pH of infants supplemented with FOS/GOS were closer to that of breast-fed infants than that of infants receiving standard formula. Infants with eczema by 18 months showed discordant development of bacterial genera of *Enterobacteriaceae* and *Parabacteroides* species in the first 26 weeks, as well as decreased acquisition of *Eubacterium* and *Anaerostipes* species, supported by increased lactate and decreased butyrate levels.
Randomized, controlled, double-blind trial with a 48 weeks intervention period from birth to 48 weeks of age.	[107]	AD outcomes at 36 weeks and 48 weeks of life. Common infections outcomes at 48 and 96 weeks of life. Relationship among early nutrition, gut microbiota and clinical outcomes.	Term infants at risk of atopy due to parental history of allergic diseases.	*n* = 201 EHF formula with added GOS: PDX (0.4 g/100 mL).	*n* = 199 standard formula	The cumulative number of infants with at least one episode of AD was not statistically different between infants supplemented or not with GOS/PDX at 36 weeks, 48 weeks and 96 weeks. The number of RIs episodes until 48 weeks were lower in infants supplemented with GOS/PDX than in infants fed with standard formula (*p* = 0.023) but no longer detected at 96 weeks. *Bifidobacteria* and *Clostridium* cluster I colonization increased over time in the supplemented group but decreased in the control groups.

(AD : atopic dermatitis ; EHF : extensively hydrolysed formulas ; CM : cow’s milk ; SCORAD: Scoring atopic dermatitis, PHF: partially hydrolysed formulas, pAOS: pectin hydrolysate-derived acidic oligosaccharides; PDX: polydextrose).

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
