# Peer review of "Prebiotics: Mechanisms and Preventive Effects in Allergy"

_nutrients, 2019, doi:10.3390/nu11081841_

Round 1

Reviewer 1 Report

Dear authors, I have carefuly reviewed your review paper 'Prebiotics: mechanism and preventive effects in allergy'. I found this a very complete overview of the preclinical and clinical evidence on prebiotics and allergy prevention and easy readible. I will only have some minor remarks.

One overall comment is about the abbreviations, many are not given (or full description is not given at first mention), please make sure this is correctly provided throughout the document (Treg, HAT, HDAC, TNF, PPARgamma, TLR and many more)

Some specific comments:

- Introduction: line 51-56 describes dietary changes (processing, reduced fiber intake etc) that may contribute to microbiota profiles, I miss references for this.

- First generation of prebiotics: line 84. An elaborate explanantion is given for prebiotics and fermentation by beneficial bacteria. Then a statements is made: 'The benefits of prevbiotics are not limited to gut, they can also act systemically.' Although there is no official description for this systematic action, this needs here a more elaborate explanation on what and how.

- Table 1. I found this table more difficult to read then the other tables, and I don't understand the bullet points in the table. Please change the layout. Below the table is written 'In blood : proven prebiotics.' It is not clear what this means.

- Second generation of prebiotics: line 120 {23-25]an, should this be [23-25] and?

- Human milk oligosaccharides: line 131 mentions >200 different HMOs structures and line 160 mentions >150 different HMO structures. What is the correct number? Use consistently.

- Figure 3. Is the figure legend from line 172-206? It now seems as a textual part of the review and not a figure legend. For the legend to be more clear, make another line when starting with B. and C. now very unclear since the text is so long

- Intestinal epithelial cells: line 324, replace 'symbiotics' with 'synbiotics' (when Gibson introduced the concept of prebiotics combined with probiotics to form what he termed as Synbiotics (DeVrese and Schrezenmeir 2008))

- Skin epithelial cells: line 379-383 in conlcusion,...... 'Knowing that skin is an important route for allergen sensitization in AD infants' is missing a reference, please add.

- Prebiotic supplementation on (I would prefer 'in') adult mice: line 446 Dietary intake of FOS during the development of food allergy were also able..... replace 'were' with 'was'

Author Response

Dear Editor and Reviewers,

Ref: Nutrients- 559847

Thank you for the opportunity to revise our review: Prebiotics: mechanisms and preventive effects in allergy. We have addressed the comments provided by the reviewer 1 as per detailed below and in the revised manuscript as indicated by track changes marked.

COMMENTS FROM REVIEWER 1.

Dear authors, I have carefuly reviewed your review paper 'Prebiotics: mechanism and preventive effects in allergy'. I found this a very complete overview of the preclinical and clinical evidence on prebiotics and allergy prevention and easy readible. I will only have some minor remarks.

One overall comment is about the abbreviations, many are not given (or full description is not given at first mention), please make sure this is correctly provided throughout the document (Treg, HAT, HDAC, TNF, PPARgamma, TLR and many more)

RESPONSE : We checked all the abbreviations and described those mentioned the first time.

Some specific comments:

- Introduction: line 51-56 describes dietary changes (processing, reduced fiber intake etc) that may contribute to microbiota profiles, I miss references for this.

REPONSE : We added this reference : « Benoit Chassaing, Matam Vijay-Kumar, and  Andrew T. Gewirtz. How diet can impact gut microbiota to promote or endanger Health. Curr Opin Gastroenterol. 2017 November ; 33(6): 417–421. doi:10.1097/MOG.0000000000000401. »

- First generation of prebiotics: line 84. An elaborate explanantion is given for prebiotics and fermentation by beneficial bacteria. Then a statements is made: 'The benefits of prevbiotics are not limited to gut, they can also act systemically.' Although there is no official description for this systematic action, this needs here a more elaborate explanation on what and how.

REPONSE : We added a more elaborate explanation : « Indeed, new original studies demonstrate that prebiotics can also modulate immune system and facilitate many biologic processes, including infection prevention and the improvement of mood and memory ».

- Table 1. I found this table more difficult to read then the other tables, and I don't understand the bullet points in the table. Please change the layout. Below the table is written 'In blood : proven prebiotics.' It is not clear what this means.

REPONSE : We changed the table 1 layout.

- Second generation of prebiotics: line 120 {23-25]an, should this be [23-25] and?

RESPONSE : We made the modifications in the text.

- Human milk oligosaccharides: line 131 mentions >200 different HMOs structures and line 160 mentions >150 different HMO structures. What is the correct number? Use consistently.

REPONSE : The more recent data on HMO structures identifed more than 150 different HMO. So, we decided to keep 150.

- Figure 3. Is the figure legend from line 172-206? It now seems as a textual part of the review and not a figure legend. For the legend to be more clear, make another line when starting with B. and C. now very unclear since the text is so long

REPONSE : Yes, the figure legend is from line 172 to 206. We made another line when starting with B. and C. We also decided to make parenthesis.

- Intestinal epithelial cells: line 324, replace 'symbiotics' with 'synbiotics' (when Gibson introduced the concept of prebiotics combined with probiotics to form what he termed as Synbiotics (DeVrese and Schrezenmeir 2008)

REPONSE : Line 324, we replaced 'symbiotics' by 'synbiotics'.

- Skin epithelial cells: line 379-383 in conlcusion,...... 'Knowing that skin is an important route for allergen sensitization in AD infants' is missing a reference, please add.

RESPONSE : We adeed this reference : « Anna R. Smith, George Knaysi, Jeffrey M. Wilson, and Julia A. Wisniewski. The Skin as a Route of Allergen Exposure: Part I. Immune Components and Mechanisms. Curr Allergy Asthma Rep. 2017 January ; 17(1): 6. doi:10.1007/s11882-017-0674-5. »

- Prebiotic supplementation on (I would prefer 'in') adult mice: line 446 Dietary intake of FOS during the development of food allergy were also able..... replace 'were' with 'was'

RESPONSE : We made the modifications.

Reviewer 2 Report

This is a very nice comprehensive review of the subject matter, the conclusions drawn are reasonable and reflect the current state of the science.

Author Response

Dear Editor and Reviewers,

Ref: Nutrients- 559847

Thank you for the opportunity to revise our review: Prebiotics: mechanisms and preventive effects in allergy. We have addressed the comments provided by the reviewer 1 as per detailed below and in the revised manuscript as indicated by track changes marked.

COMMENTS FROM REVIEWER 2.

This is a very nice comprehensive review of the subject matter, the conclusions drawn are reasonable and reflect the current state of the science.

RESPONSE : We thank you for your comments.